# In Vitro Fertilisation of Mouse Oocytes in L-Proline and L-Pipecolic Acid Improves Subsequent Development

**DOI:** 10.3390/cells10061352

**Published:** 2021-05-29

**Authors:** Tamara Treleaven, Madeleine L.M. Hardy, Michelle Guttman-Jones, Michael B. Morris, Margot L. Day

**Affiliations:** Physiology, School of Medical Sciences, Faculty of Medicine and Health, University of Sydney, Sydney, NSW 2006, Australia; tprz7203@uni.sydney.edu.au (T.T.); mhar7073@uni.sydney.edu.au (M.L.M.H.); drmguttmanjones@gmail.com (M.G.-J.); m.morris@sydney.edu.au (M.B.M.)

**Keywords:** oocyte, amino acid transporters, proline, pipecolic acid, in vitro fertilisation

## Abstract

Exposure of oocytes to specific amino acids during in vitro fertilisation (IVF) improves preimplantation embryo development. Embryos fertilised in medium with proline and its homologue pipecolic acid showed increased blastocyst formation and inner cell mass cell numbers compared to embryos fertilised in medium containing no amino acids, betaine, glycine, or histidine. The beneficial effect of proline was prevented by the addition of excess betaine, glycine, and histidine, indicating competitive inhibition of transport-mediated uptake. Expression of transporters of proline in oocytes was investigated by measuring the rate of uptake of radiolabelled proline in the presence of unlabelled amino acids. Three transporters were identified, one that was sodium-dependent, PROT (SLC6A7), and two others that were sodium-independent, PAT1 (SLC36A1) and PAT2 (SLC36A2). Immunofluorescent staining showed localisation of PROT in intracellular vesicles and limited expression in the plasma membrane, while PAT1 and PAT2 were both expressed in the plasma membrane. Proline and pipecolic acid reduced mitochondrial activity and reactive oxygen species in oocytes, and this may be responsible for their beneficial effect. Overall, our results indicate the importance of inclusion of specific amino acids in IVF medium and that consideration should be given to whether the addition of multiple amino acids prevents the action of beneficial amino acids.

## 1. Introduction

The presence of individual and groups of amino acids in embryo culture medium impacts preimplantation embryo development [1,2,3,4,5,6,7,8]. Much less is known, however, about the impact of individual amino acids on future development when they are exogenously added to oocytes during in vitro fertilisation (IVF). The addition of specific amino acids or groups of amino acids to the IVF medium can improve various aspects of subsequent development, including the percentage of embryos that reach the blastocyst stage and then hatch [9,10], indicating that amino acids can enter the oocyte in sufficient quantities during the period of fertilisation to exert effects on later embryo development.

The amino acid L-proline (Pro) acts in a growth-factor-like manner to stimulate pre-implantation development [7]. When zygotes are cultured at low density (1 embryo/100 µL, to eliminate the action of embryo-derived growth factors), the addition of Pro increases both the number of embryos reaching the blastocyst stage and their hatching by stimulating development within the late two-cell to eight-cell stages [7]. The addition of Pro also promotes differentiation and neural lineage commitment in embryonic stem cell (ESC) models of post-implantation development [11,12].

The mechanisms by which Pro stimulates pre- and post-implantation development are only partially understood. In embryos cultured at low density, Pro stimulates nuclear translocation of p-Akt^S473^ and p-ERK1/2^T202/Y204^, and improvement in development is dependent on mTORC1 signalling (rapamycin sensitive) [3]. Activation of these signalling pathways is also involved in the differentiation of ESCs [11,13,14]. Cellular handling of the production of this conditional amino acid is also involved: ESCs limit the endogenous production of Pro via a mechanism involving the amino acid response pathway, to the extent that differentiation is suppressed and ESC self-renewal maintained [15]. The exogenous addition of Pro overwhelms this suppressive mechanism. Pro also promotes changes to the epigenetic landscape of ESCs via enhanced histone methylation (at H3K9 and H3K36). The mechanism may involve direct or indirect inhibition of the Jumonji domain demethylases, which help control methylation status at these sites [16].

Attention has also turned to Pro’s unique metabolism as a source of mechanisms controlling developmental fate [13,14,16,17,18]. Pro is metabolised via the Pro cycle to pyrroline-5-carboxylic acid (P5C) by proline oxidase (POX), and then to glutamate semialdehyde before being converted to glutamate (Glu) and α-ketoglutarate [17,19]. Thus, Pro metabolism can provide a number of intermediates important for cellular function, including the production of ATP via both α-ketoglutarate production and Pro-derived high-energy electrons entering the electron transport system [17].

Pro also has the ability to scavenge intracellular reactive oxygen species (ROS) [20]. Embryos with low metabolism, low glycolytic activity, low amino acid turnover, and high antioxidant levels are hypothesised to be the most viable [21,22,23,24], most likely because the embryo expends less energy and has lower production and leakage of superoxide free radicals from mitochondria [21,25,26]. H_2_O_2_ is another source of ROS, produced by superoxide dismutase conversion of the superoxide radical, especially during the G_2_/M phase of cleavage-stage embryos [27,28,29]. High concentrations of ROS are detrimental to oocyte quality: Oocytes exposed to high levels of ROS in vivo have lower fertilisation rates, produce lower quality embryos, and present increased embryo arrest at the two-cell stage in the mouse [28,30].

The transport of Pro into cells occurs via specific amino acid transporters, specifically neutral amino acid transport systems, and are further characterised by their gene sequence homology in the solute carrier transporter (SLC) family. In the mouse oocyte, a number of amino acid transport systems have been identified (L, b^0,+^, ASC, asc, B^0^, Gly, beta), one or more of which may be responsible for the uptake of Pro into the oocyte [31]. Uptake of Pro into cumulus cells surrounding the oocyte also occurs via y+LAT2 (SLC7A6), and thus Pro is transferred to the oocyte via gap junctions in the cumulus oocyte complex [32,33]. mRNA for the Na^+^-dependent Pro transporter PROT (SLC6A7) and the Na^+^/Cl^−^-dependent betaine/proline transporter SIT1 (SLC6A20) are present in the mouse oocyte, but it is not known if the transporters themselves are expressed and actively take up Pro [34]. After fertilisation, large amounts of Pro are taken up by SIT1, which is active in the zygote and two-cell stages [35]. Pro uptake in zygotes also involves an unidentified Na^+^-dependent betaine-resistant transporter [34,35]. In ESCs, Pro uptake is via the System A Na^+^-dependent SNAT2 transporter (SLC38A2) [14]. System A transporters are also the major transport system in the inner cell mass of blastocysts, but it is not known if SNAT2 is the principal contributor [36]. The different amino acid requirements at specific stages of oocyte and embryonic development reflect the dynamic physiology of the preimplantation embryo, including changes in metabolism and amino acid transporter expression.

L-pipecolic acid (PA) is a product of lysine metabolism and a homologue of Pro, having a six-membered piperidine ring rather the five-membered pyrrolidine ring of Pro [20]. Both PA and Pro undergo oxidation in mitochondria and promote cell survival during oxidative stress [37]. It is not known if there is a transporter for PA in oocytes, but in somatic cells PA is transported by the high-affinity Pro transporter PROT [38,39,40] and SIT1 [41,42]. We hypothesised that because PA is structurally similar to Pro it would be taken up into the oocyte and be either metabolised or act as a ROS scavenger in a similar way to Pro, and thereby improve later embryo development [37,43].

This study investigated the effect of the addition of specific amino acids to IVF medium on subsequent preimplantation embryo development, with a particular focus on Pro and its analogue PA. We also examined the effect of molar excess of specific amino acids on uptake of Pro to determine the transporter(s) responsible for uptake. Finally, we investigated metabolic mechanisms by which Pro and PA may impact oocyte development viability after IVF. Overall, this study shows that Pro and PA can improve preimplantation embryo development when added to IVF medium by reducing mitochondrial activity and ROS in the oocyte.

## 2. Materials and Methods

### 2.1. Animals (Mus Musculus)

Quackenbush Swiss (QS) mice (Animal Resource Centre, Perth and Lab Animal Services, University of Sydney) were housed under 12 h light:12 h dark conditions as described previously [44]. Experiments were conducted in accordance with the Australian Code of Practice for the Care and Use of Animals for Scientific Purposes and approved by the University of Sydney Animal Ethics Committee as required by the NSW Animal Research Act (protocols 5583 and 824).

### 2.2. Isolation of Oocytes and Zygotes Fertilised In Vivo

Superovulation of female mice aged 4–6 weeks was achieved by intraperitoneal injection with 10 IU pregnant mares’ serum gonadotropin (PMS; Intervet, Vic, Australia) followed after 48 ± 2 h by injection with 10 IU human chronic gonadotropin (hCG; Intervet). To obtain zygotes fertilised in vivo, female mice were paired overnight with individually housed QS male mice (2–8 months of age) immediately following hCG injection. Mating was confirmed the following day by the presence of a vaginal plug. Mice were sacrificed by cervical dislocation at either 13–15 h post-hCG to obtain oocytes or 22 h post-hCG to obtain zygotes. Oocytes and zygotes were isolated into HEPES-buffered modified human tubal fluid (Hepes-modHTF) in which the concentration of NaCl was adjusted to 85 mM to give an osmolality of 270 mOsm/kg [7]. Cumulus cells were removed from zygotes and oocytes by treatment with 1 mg/mL hyaluronidase in HEPES-modHTF and then washed 3 times in HEPES-modHTF to remove remaining cumulus cells.

### 2.3. In Vitro Fertilisation of Oocytes and Culture of Embryos

In vitro fertilisation was performed in modified Whittingham’s medium [45,46] which contained (in mM) NaCl (99.3), KCl (2.7), NaH_2_PO_4_ (0.36), MgCl_2_.6H_2_O (0.5), NaHCO_3_ (25), Na pyruvate (0.5), glucose (5.5), CaCl_2_.2H_2_O (1.8), Na lactate (18.7), and 30 mg/mL BSA. At least 4 cumulus masses containing oocytes were added to 500 µL pre-equilibrated (5% CO_2_ at 37 °C) Whittingham’s medium, containing the appropriate amino acid(s), overlayed with mineral oil (Sigma-Aldrich, NSW, Australia). Treatment groups were (i) no amino acids (no AA), and (ii) 0.4 mM L-proline (Pro) ± 5 mM His, Gly, betaine (Bet) or PA (as potential competitive inhibitors of the uptake of Pro). The concentration of Pro used in these experiments was based on the physiological concentration of fluid in the reproductive tract [47] and was shown to improve embryo development [7].

Sperm was isolated from the epididymis of a male QS mouse of proven fertility. An incision was made in the efferent ducts between the testis and the epididymis, and at the inferior end of the caudis epididymis. The dissected tissue was immediately washed, and blood vessels drained in PBS. The epididymides were placed into 500 µL Whittingham’s medium at 37 °C and 5% CO_2_ and the sperm gently squeezed out of the epididymides using forceps and allowed to capacitate for 1.5 h. At 13.5 h post-hCG, 10^6^ sperm were added to each drop containing cumulus masses and incubated together for 3 h, after which fertilisation was determined by the presence of a polar body. Zygotes were washed through at least 3 drops of HEPES mod-HTF in the absence of amino acids and then cultured to the blastocyst stage (120 h) at low density (1 embryo/100 µL) in mod-HTF + 0.3 mg/mL BSA, containing no amino acids, at 37 °C in 5% CO_2_ under mineral oil [7]. Embryos were scored for their developmental stage every 24 h. In vivo fertilised zygotes were also cultured under these conditions as a control.

### 2.4. Measurement of L-[^3^H]-Pro Uptake in Oocytes

Oocytes were incubated in 20 µL assay medium consisting of 1 µM L-[^3^H]-Pro (L-[2,3,4,5-^3^H]-proline; 1 mCi/mL; Perkin-Elmer, Vic, Australia, NET483001MC) in HEPES mod-HTF ± molar excess of individual unlabelled amino acids (5 mM for all amino acids except Pro, which was used at 0.4 mM). Unless otherwise stated, the L-isomer was used for all unlabelled amino acids and purchased from Sigma Aldrich, except L-Lys (Chem-Impex, IL, USA). In experiments using Na^+^-free HEPES mod-HTF, the Na^+^ was replaced with N-methyl-D-glucamine (NMDG) at 85 mM. At this low concentration NMDG is not known to affect Pro uptake by Na^+^-independent transporters.

An initial time course experiment was performed in which oocytes were incubated in L-[^3^H]-Pro for up to 120 min and samples collected at 20-min intervals to show that the uptake of L-[^3^H]-Pro remained linear over this time course (data not shown). In all subsequent experiments, oocytes in groups of 4 were incubated for 100 min at 37 °C in each treatment and then washed through at least 3 drops of cold (4 °C) HEPES mod-HTF, aspirated, and placed onto a gridded filter mat (Perkin-Elmer, Vic, Australia, #1450-421). The mat was placed inside a plastic sleeve and 4 mL ULTIMA GOLD scintillation fluid (Perkin-Elmer, Vic, Australia, #6013371) was added. The plastic sleeve was placed in a MicroBeta TriLux Plate Counter (Perkin-Elmer, Vic, Australia) and each sample was counted for 30 min. To determine the rate of L-[^3^H]-Pro uptake (fmol min^−^^1^ oocyte^−^^1^), a standard curve was created for each experiment from standards serially diluted from a stock of 1 µM L-[^3^H]-Pro in HEPES mod-HTF and fitted by linear regression.

### 2.5. Immunostaining and Confocal Microscopy

Oocytes were fixed in 4% paraformaldehyde (PFA) for 30 min at room temperature, then washed 3 times in PBS + 1 mg/mL polyvinyl acid (PBS + PVA; Sigma-Aldrich, NSW, Australia), and permeabilised with PBS + PVA + 0.3% Triton X-100 for 30 min at room temperature. The oocytes were blocked by incubation in PBS + PVA + 0.1% Tween-20 + 0.7% BSA for 30 min at room temperature. Antibodies used were rabbit anti-SLC6A7/PROT (GeneTex, CA, USA #GTX51242), rabbit anti-SLC36A1/PAT1 [48], rabbit anti-SLC36A2/PAT2 [48], and Alexa Fluor 488-coupled goat anti-rabbit IgG (Invitrogen, Vic, Australia). Primary antibodies were diluted 1:500, and the secondary antibody 1:200, in PBS + PVA + 0.1% Tween-20 + 0.7% BSA. Samples were incubated in primary antibody for 2 h at room temperature or overnight at 4 °C and then washed 3 times in PBS + PVA + 0.1% Tween-20 + 0.7% BSA. Samples were incubated in the dark in secondary antibody for 1 h at room temperature and then washed 3 times in PBS + PVA + 0.1% Tween-20 + 0.7% BSA before being mounted in 3 μL Vectashield containing 1.5 µg/mL DAPI (Vector Laboratories, CA, USA) to visualise nuclei. Oocytes were imaged using confocal microscopy (LSM Meta 800, Carl Zeiss, Oberkochen, Germany) using 405 nm and 488 nm lasers and a 40× objective. Images were prepared using Fiji by Image J.

For analysis of cell numbers, blastocysts were fixed and permeabilised as described above and mounted in VECTASHIELD containing DAPI. Nuclei were visualised by confocal microscopy using a 405 nm laser. A Z-stack was taken through each blastocyst at 2.5 µm slices to allow for each nucleus to be imaged at least once. Cells in the inner cell mass and trophectoderm were counted manually using Image J. Cell numbers from each blastocyst were averaged.

### 2.6. Measurement of Mitochondrial Activity and Reactive Oxygen Species

Oocytes were cultured for 3 h in Whittingham’s medium containing 0.4 mM Pro ± 5 mM PA Gly, Bet or His. Oocytes were then transferred to Whittingham’s medium containing the above amino acid combinations to which 100 nM tetramethylrhodamine methyl ester (TMRM) and 10 µM 2′,7′-dichlorofluorescin diacetate (DCFDA) were added. These were incubated for 30 min at 37 °C in 5% CO_2_ to allow uptake of the dyes, then washed twice through 2 drops of Whittingham’s medium, containing the amino acid combinations above, to remove excess dye. A 35 mm glass-bottomed petri dish was prepared containing 10 μL drops of medium (under mineral oil) containing each of the treatment conditions ±5 µM carbonyl cyanide-*4* (trifluoromethoxy) phenylhydrazone (FCCP) as a positive control for mitochondrial activity, or ±60 µM hydrogen peroxide (H_2_O_2_) as a positive control for ROS, pre-equilibrated to 37 °C and 5% CO_2_. Oocytes were added to drops and imaged using confocal microscopy (Carl Zeiss, Oberkochen, Germany, LSM Meta 800), with the chamber set to 37 °C and 5% CO_2_ for live-cell imaging, using a 20× objective and 524 nm and 488 nm lasers. All oocytes were imaged within 4 h of isolation, which was equivalent to the timeline of amino acid treatment in the IVF experiments. Images were analysed using Fiji by Image J to obtain the integrated density of the stain, and the corrected total cell fluorescence was calculated using the following formula: Corrected total cell fluorescence = integrated density—(background fluorescence × mean area) [49].

### 2.7. Statistical Analyses

Oocyte/embryo culture experiments were performed a minimum of 3 times with at least 10 oocytes/embryos per treatment group. Development in the different treatment groups was compared by calculating the percentage of embryos that developed to the particular cell stage. Differences between groups were determined using 1-way ANOVA and Tukey’s or Dunnett’s multiple comparisons test as indicated in the figure legends.

L-[^3^H]-Pro uptake experiments were performed at least 3 times with 4 embryos per treatment group. The uptake of L-[^3^H]-Pro for each treatment was averaged and the difference between groups compared by 1-way ANOVA and Tukey’s multiple comparisons test.

For fluorescence imaging, corrected total cell fluorescence was averaged from 3 independent experiments with a total of 28–71 embryos per treatment, and differences between treatments were determined by one-way ANOVA and Tukey’s multiple comparisons test. Data analysis was performed using GraphPad Prism v7.

## 3. Results

### 3.1. IVF in the Presence of Pro and PA Improves Subsequent Embryo Development

Oocytes were fertilised for 3 h in the presence or absence of various amino acids, and then subsequently allowed to develop in vitro in the absence of any amino acids. In vitro development was performed at low density (1 embryo/100 µL) and, therefore, in the absence of autocrine support. The proportion of oocytes fertilised was not significantly affected by any amino acid treatment (No AA 45%, Pro 47%, PA 50%, Gly 45%, His 39%, Pro + PA 45%, Pro + Gly 45%, His + Pro 39%). However, addition of 0.4 mM Pro to the medium during IVF improved subsequent development as compared to oocytes fertilised in the absence of amino acids (Figure 1A–D). In particular, the percentage of embryos that had compacted at 72 h and developed to the blastocyst stage at 120 h following IVF in Pro increased compared to oocytes fertilised in the absence of AA. The presence of Pro during IVF also increased the number of ICM cells, with the number reaching levels seen in blastocysts developed from in vivo fertilised oocytes (Figure 1E). IVF in the presence of Pro had no effect on trophectoderm cell number in blastocysts (Figure 1F).

PA is transported by some of the transporters of Pro and may, therefore, compete for and inhibit cellular uptake of Pro by those transporters. Thus, it was hypothesised that the presence of PA in molar excess (5 mM) over Pro (0.4 mM) during IVF would inhibit the effects of Pro. Instead, addition of PA alone to medium during IVF produced results similar to those observed for Pro alone (Figure 1A,E,F), increasing percent development to the compacted and blastocyst stages and increasing ICM cell numbers in blastocysts. The combination of PA with Pro during IVF had no additive or synergistic effects (Figure 1A,E).

### 3.2. Gly, Bet and His Inhibit Pro-Mediated Improvement in Development after IVF

The effect of other potential competitors of Pro uptake during IVF was also examined. When 5 mM Gly, Bet, or His were added individually to the IVF medium, none had any effect on embryo development compared to IVF undertaken in the absence of added amino acids (Figure 1B–F). However, each of these amino acids inhibited the effects observed with Pro (Figure 1B–E), indicating competition for transport of Pro into the oocyte.

### 3.3. Transport of Pro into Oocytes Has Na^+^-Dependent and Na^+^-Independent Components

A number of amino acid transporters are expressed in the mouse oocyte [31]. To determine which of these transporters is responsible for the transport of Pro, we examined the rate of 1 µM L-[^3^H]-Pro uptake in the presence of a range of unlabelled amino acids in molar excess. Excess Pro itself was used to show the proportion of L-[^3^H]-Pro uptake that was saturable (i.e., due to a transporter). Common L-amino acids and D-Pro were chosen based on the preferred substrates of known Pro transporters [38]. Trp was chosen as a known inhibitor of specific Na^+^-independent transporters [50,51].

We found uptake of Pro involves both Na^+^-dependent (Figure 2A) and Na^+^-independent (Figure 2B) transport mechanisms. In Na^+^-containing medium, Pro, D-Pro, PA, Gly, Ala, Bet, His, and sarcosine (Sar) all inhibited the uptake of L-[^3^H]-Pro (*p* < 0.01). Lys, Leu, Ser, and Gln did not decrease the L-[^3^H]-Pro rate of uptake. In Na^+^-free medium, D-Pro, Gly, Trp, Bet, Pro, and PA inhibited the L-[^3^H]-Pro rate of uptake (*p* < 0.05), whereas His and Leu had no effect.

### 3.4. PROT and PAT1/2 Are Expressed in Oocytes

We investigated the expression of three Pro transporters with similar amino acid uptake profiles to what we observed in the oocyte. Expression of PROT, PAT1, and PAT2 was examined by immunofluorescent staining. SLC6A7 (PROT) displayed puncta across the cytoplasm as well as some plasma-membrane staining over the location of the metaphase II spindle (Figure 3). This indicates that PROT is involved in both membrane and vesicular transport of Pro. It is likely that expression of PROT in vesicles contributes to Pro accumulation and/or signalling processes important for oocyte competency. For SLC36A1 (PAT1) and SLC36A2 (PAT2), staining was detected in the plasma membrane as well as in sub-cortical puncta (Figure 3). No staining was detected with the IgG control.

### 3.5. Pro and PA Significantly Reduce Mitochondrial Activity in Oocytes

Reduced metabolism has been linked to improved embryo viability [21]. We hypothesised that the presence of Pro or PA during IVF would improve embryo development by decreasing metabolism in the oocyte. Mitochondrial membrane potential is used as a measure of mitochondrial activity and therefore the metabolic activity of cells [21]. Thus, to determine if Pro, PA, or other amino acids were affecting metabolic activity, we examined mitochondrial membrane potential using the potentiometric indicator TMRM in oocytes cultured in medium containing specific amino acids for 3 h. The presence of Pro or PA in the medium reduced TMRM fluorescence compared to the no AA control (Figure 4A,B). The presence of both Pro and PA did not have an additive effect. Gly and His alone had no effect on TMRM fluorescence, while Bet reduced TMRM fluorescence, but not to the same extent as Pro or PA. Gly, Bet, and His all prevented the Pro-induced decrease in TMRM fluorescence. FCCP uncouples mitochondrial oxidative phosphorylation by interfering with the transport of protons across the mitochondrial membrane. As expected, FCCP decreased TMRM fluorescence and mitochondrial activity (Figure 4D).

### 3.6. Pro and PA Significantly Reduce Reactive Oxygen Species in Oocytes

ROS accumulation is detrimental to fertilisation and embryo development [28,30]. Since Pro and PA can act as ROS scavengers [20,37], we investigated whether they were performing this role in the oocyte. ROS were measured in oocytes using DCFDA, which is oxidized by ROS, increasing fluorescence of the dye [52]. The presence of Pro or PA in the medium reduced DCFDA fluorescence in oocytes reflecting a decrease in ROS in comparison to the no AA control (Figure 4A,C). The presence of Gly, Bet, and His alone did not affect DCFDA fluorescence but prevented the Pro-dependent decrease in fluorescence. The positive control, H_2_O_2_, caused a large increase in ROS, as expected (Figure 4E).

## 4. Discussion

In this study, we exposed oocytes to individual amino acids in fertilisation medium to determine if any subsequently improved preimplantation development. In the reproductive tract, oocytes are bathed in oviductal fluid, which contains amino acids and other factors that contribute to normal sperm capacitation, oocyte fertilisation, and preimplantation development. When the oocyte is fertilised outside of its physiological environment, its development potential is significantly impacted by the components of medium [53]. The amino acid composition of commercially available media varies [54].Although it is known that individual amino acids can benefit embryo development [7,55,56], little is known about how the presence of specific amino acids during fertilisation might impact on later embryo development, and this was therefore the focus of this study.

Fertilisation of oocytes was not significantly affected by supplementing fertilisation medium with any of the amino acids tested (Bet, Gly, His, Pro, and PA). Similarly, the addition of 19 common amino acids together does not increase the proportion of oocytes fertilised when IVF is performed on bovine oocytes [57]. This suggests that amino acids do not impact either the sperm–egg interaction or fertilisation processes occurring after sperm–oocyte fusion. However, the presence of Pro or PA during IVF (followed by their removal) resulted in an increase in the percentage of compacted embryos and blastocysts and inner cell mass cell numbers, suggesting improved embryo viability. IVF in the presence of Bet, Gly, or His did not improve embryo development by these criteria, but they did inhibit the improvements mediated by Pro or PA, suggesting that Bet, Gly, and His compete for uptake of Pro by the same amino acid transporter. These results are consistent with a growing body of evidence showing that competitive inhibition of uptake prevents the beneficial effects that selected amino acids have on various stages of the development process [7,8,11,14,34,35].

In somatic cells there are multiple transporters for Pro including SIT1, PAT1 and 2, PROT, SNAT2, B^0^AT1 and 2, NTT4, ASCT1, y+LAT2, and GLYT1 [31,34,38,50,58,59,60,61]. Of these, SIT1 [34,35], GLYT1 [1,62,63], B^0^AT2 [64], PROT [35], and SNAT2 [65] have been described in mouse embryo developmental stages after fertilisation. In oocytes, both Na^+^-dependent (~50% of transport) and Na^+^-independent (~30% of transport) Pro transport were observed. The Bet/Pro transporter, SIT1, can be eliminated as a candidate for Pro uptake since it is not active until after fertilisation [34]. Similarly, y+LAT2 can be eliminated because it is only expressed in cumulus cells [32,33]. Using competitors for L-[^3^H]-Pro uptake, it was also possible to eliminate a number of other transporters as candidates for Pro uptake in the oocyte: B^0^AT1, B^0^AT2, and NTT4 can be eliminated because Leu did not prevent Na^+^-dependent Pro uptake [14,58,60,61]. We can also eliminate the ASC/asc transporter family and SNAT2 since Ser did not reduce Pro uptake [14,31,66]. GLYT1 is located in the oocyte plasma membrane and actively transports Gly at fertilisation and throughout preimplantation development [1,62,67]. As GLYT1 substrates Gly and Sar both reduced Pro uptake, we cannot rule out the possibility that GLYT1 takes up Pro and PA in oocytes. However, its weak affinity for Pro makes it unlikely to be the main contributor [63]. It should also be noted that some transporters in oocytes and early embryos have characteristics that differ from the corresponding cloned transporters [64,65]. However, based on the available evidence, this leaves PROT, which is Na^+^-dependent, and PAT1 and PAT2, which are Na^+^-independent, as candidate transporters for Pro in the oocyte [68].

PROT is a good candidate for Pro transport in the oocyte as it is a high-affinity Na^+^-dependent Pro transporter which also transports PA, Sar, and His [69]. Immunostaining for PROT in oocytes showed localisation in cytoplasmic vesicles and limited plasma membrane staining concentrated near the MII chromosomes. In the central nervous system PROT takes up Pro for normal brain function, memory, and locomotor activity [67]. PROT-null mice exhibit brain dysfunction but have no reported fertility or embryological defects suggesting that there is Pro transporter redundancy [70]. It is also likely that compensatory mechanisms are activated when important genes are knocked out that are needed for normal development, as may be the case for PROT. For the Na^+^-independent transporters PAT1 and PAT2, the substrates D-Pro, Gly, and Trp [50,71] all prevented Pro uptake. In addition, immunostaining for both PAT1 and PAT2 showed both are located in the plasma membrane of oocytes. Gly and Ala are found in high concentrations in oviductal fluid which would compete with the uptake of Pro, so it is unlikely that PAT1/2 are the only functional transporters of Pro in vivo [47,72]. Future studies could investigate whether the observed inhibition of Pro transport by the relatively high concentrations of Gly and Ala used in this study was competitive or non-competitive. Our data, therefore, suggest there are multiple transporters of Pro in the oocyte with overlapping substrate specificities. Under in vitro conditions, the relative activity of each of these transporters in oocytes will depend on the amino acid composition of the medium. In particular, the beneficial effects on later embryo development through the exogenous addition of Pro during fertilisation might be mitigated or eliminated through inappropriate addition of other amino acids which reduce its uptake.

Pro and PA may have an immediate impact during IVF by reducing mitochondrial activity and ROS levels in the oocyte. Pro acts as an antioxidant and cryoprotectant by reducing ROS and mitochondrial activity in vitrified mouse oocytes [73]. Amino acids such as Pro which contain a secondary amine group have a lower ionizing potential than those with only a primary amine group, enabling them to easily donate electrons and quench ROS by stabilisation of the free radical [20,74]. The structurally similar PA appears to perform the same functions, acting as a scavenger of ROS and preventing their build-up in the oocyte. Gly, Bet, and His do not have a secondary amine group and were not able to reduce ROS. Instead, all three prevented the Pro-mediated reduction in ROS due to their competitive inhibition of Pro uptake via PAT1/2 and/or GLYT1. This scavenging mechanism is one potential pathway by which Pro and PA reduce ROS in the oocyte.

Both Pro and PA reduced mitochondrial activity in the oocyte. Our data therefore support the theory that metabolically quiet oocytes and embryos have improved viability [21,22]. Metabolism of Pro by proline oxidase produces P5C, which is converted to Glu. Glu can then be either converted to α-ketoglutarate, which enters the TCA cycle, or combine with cysteine and glycine to form the antioxidant glutathione [18,75]. Since the presence of Pro caused a decrease in mitochondrial activity, our data suggest that Pro metabolism does not feed significantly into the TCA cycle. It did not feed high-energy electrons into the ETC, but instead increased glutathione (GSH) production. Pro-mediated increases in GSH would also contribute to reduced ROS [20,76], as has been shown in boar sperm [76] and in-vitro cultured mouse embryos (unpublished data). Alternately, the Pro-induced reduction in mitochondrial activity indicates reduced ETC activity, which may be the mechanism by which Pro reduces ROS. Although the exact mechanism by which Pro reduces mitochondrial activity has not yet been identified, a Pro-induced simultaneous reduction of mitochondrial activity and ROS levels has been observed in other cells [77].

PA is metabolised similarly to Pro, whereby it is broken down by pipecolate oxidase to piperideine-6-carboxylate (P6C) and then to α-aminoadipic semialdehyde [37], which can result in increased production of glutamate and thus potentially GSH [78]. As Pro and PA both improve embryo development after IVF and decrease ROS and mitochondrial activity, we suggest that increased GSH due to Pro and PA metabolism may be one of the mechanisms responsible for their beneficial effect on later embryo development.

There are, however, important differences in development when Pro is added at the time of fertilisation compared to when it is added to already fertilised oocytes [7]. In the latter case, Pro (i) first improves embryo development during cleavage stages (late two-cell to eight-cell) instead of from compaction, and (ii) has no effect on blastocyst cell numbers [7]. This implies that the Pro-mediated mechanisms required to ‘set up’ improvement in later development are importantly different between these two scenarios and are dependent on the timing of the exogenous addition of this amino acid. Nevertheless, the molecular mechanisms at play when Pro is added during fertilisation may well also include one or more of those established for Pro in related work [7,11,15,16,79]. Finally, Pro added during fertilisation in vitro increased the number of inner cell mass cells in blastocysts to the level seen for blastocysts derived from cultured zygotes fertilised in vivo. However, trophectoderm cell numbers were not similarly increased. This is consistent with mouse studies showing that IVF reduces trophoblast cell numbers and affects subsequent placental development due to alterations in gene expression including the downregulation of SLC genes [80,81]. This indicates that one or more factors are still missing in the in vitro fertilisation environment.

In conclusion, Pro and PA are beneficial to preimplantation embryo development when added to the fertilisation medium. Pro and PA reduce mitochondrial activity and ROS in oocytes. The reduction in ROS by Pro and PA may be due to one or more of a variety of mechanisms: (i) The reduction in mitochondrial activity per se, (ii) the direct scavenging of ROS by these molecules, and (iii) their metabolism leading to increased production of GSH. We suggest that these may be some of the mechanisms responsible for the beneficial effect of Pro and PA. The beneficial actions of Pro are prevented when other amino acids are present in the IVF medium due to competitive uptake by amino acid transporters. Very little attention has been given to the amino acid composition of media used during fertilisation; rather, research has focused on culture media used during embryo development. This study shows that specific amino acids present during the process of fertilisation have impacts on later development. Indeed, the composition of currently used commercial IVF media may prevent the beneficial effect of Pro by oversupplying amino acids which prevent Pro uptake. We suggest a simplified fertilisation medium that contains key elements required to drive development in vivo that is a favourable strategy for promoting oocyte competency and subsequent embryo viability in vitro.

## Figures and Tables

**Figure 1 cells-10-01352-f001:**
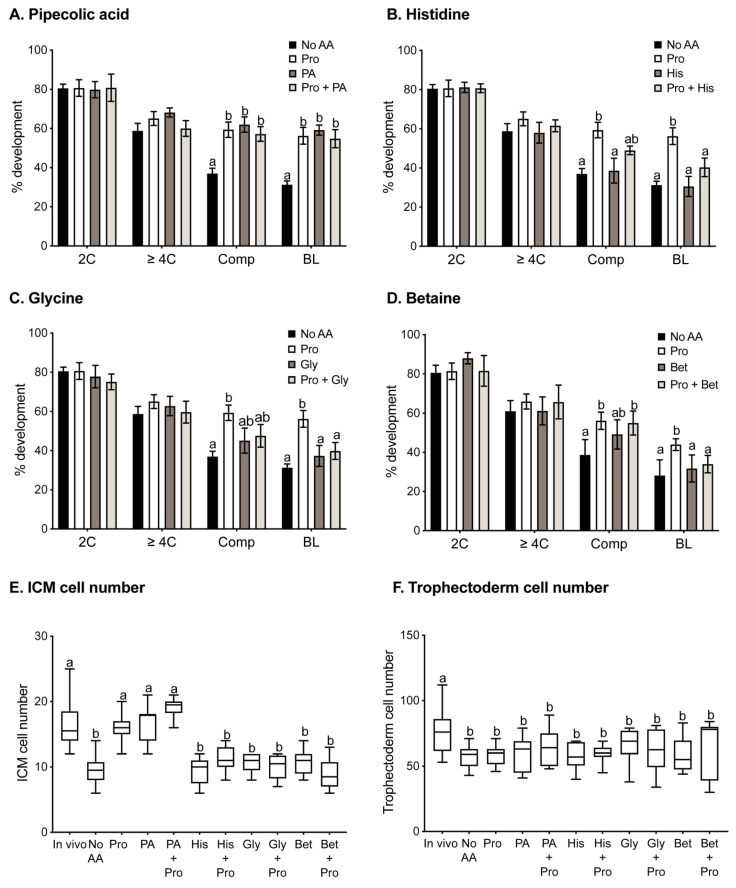
Preimplantation development following IVF in the presence of various amino acids. Fertilisation was performed in the absence of amino acids (No AA) or the presence of 0.4 mM Pro alone or 0.4 mM Pro in the presence of (**A**) 5 mM PA, (**B**) 5 mM His, (**C**) 5 mM Gly, or (**D**) 5 mM Bet. Following fertilisation, zygotes were cultured in mod-HTF in the absence of added amino acids for 120 h, and the percentages of embryos at the 2-cell (24 h), ≥4-cell (48 h), compacted (72 h), and blastocyst (120 h) stages were recorded. Bars represent mean ± SEM obtained from 3 independent experiments with a minimum of 10 zygotes per treatment group in each experiment. Blastocysts were fixed and stained with DAPI, and ICM (**E**) and trophectoderm (**F**) cell numbers were counted. Whiskers indicate highest and lowest values, and the mean is represented by the centre line in the box. Data were analysed using GraphPad Prism using 1-way ANOVA with Tukey’s multiple comparisons test (**A**–**D**) or Dunnett’s multiple comparison test (**E**,**F**). Bars sharing the same letter are not significantly different (*p* > 0.05).

**Figure 2 cells-10-01352-f002:**
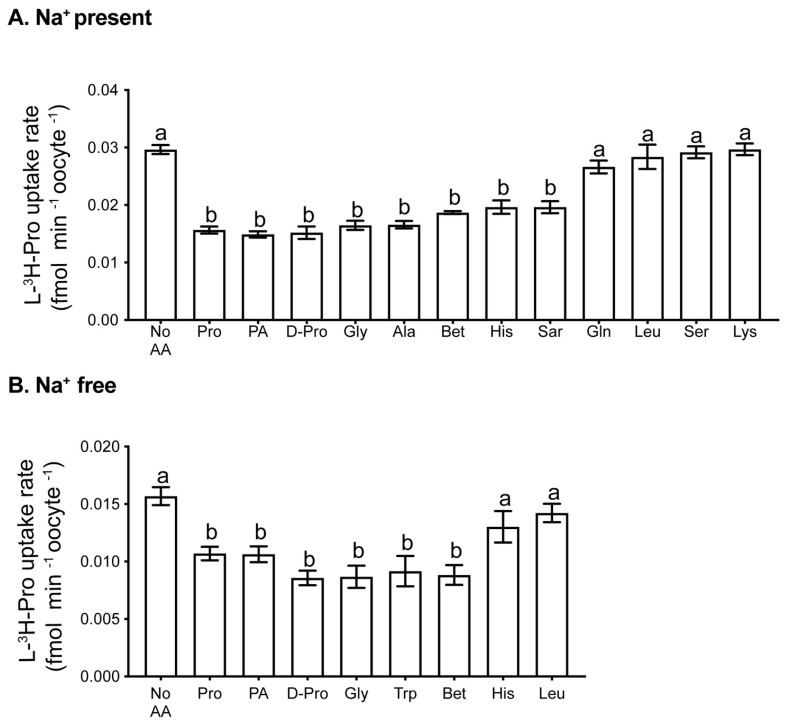
Uptake of L-[^3^H]-Pro by oocytes in the presence of other amino acids. Rate of uptake of 1 µM L-[^3^H]Pro in (**A**) mod-HTF containing Na^+^ or (**B**) Na^+^-free mod-HTF in the presence of unlabelled competing amino acids. The concentration of all competitors was 5 mM, with the exception of Pro (0.4 mM). Each bar represents the mean ± SEM from at least 3 independent experiments, each with treatments performed in triplicates of 4 oocytes. Data were analysed using 1-way ANOVA with Tukey’s post hoc test. Bars with different letters are significantly different (*p* < 0.05). Sar: sarcosine.

**Figure 3 cells-10-01352-f003:**
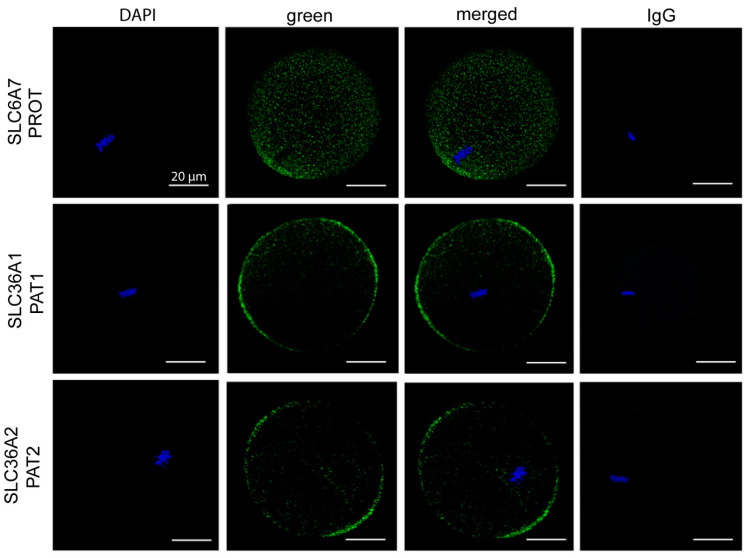
Expression of Pro transporters PROT, PAT1, and PAT2 in mouse oocytes. Oocytes were freshly isolated at 13–15 h post-hCG and exposed to Pro (0.4 mM) for 100 min, fixed, and immunostained for SLC6A7 (PROT), SLC36A1 (PAT1), and SLC36A2 (PAT2) (green). Metaphase chromosomes were counterstained with DAPI. Oocytes were analysed by confocal microscopy using a 40× objective. Images are representative of 6 oocytes taken from 3 independent experiments. A pre-immune IgG equivalent was used as a negative control for the primary antibody. Scale bar = 20 µm for all images.

**Figure 4 cells-10-01352-f004:**
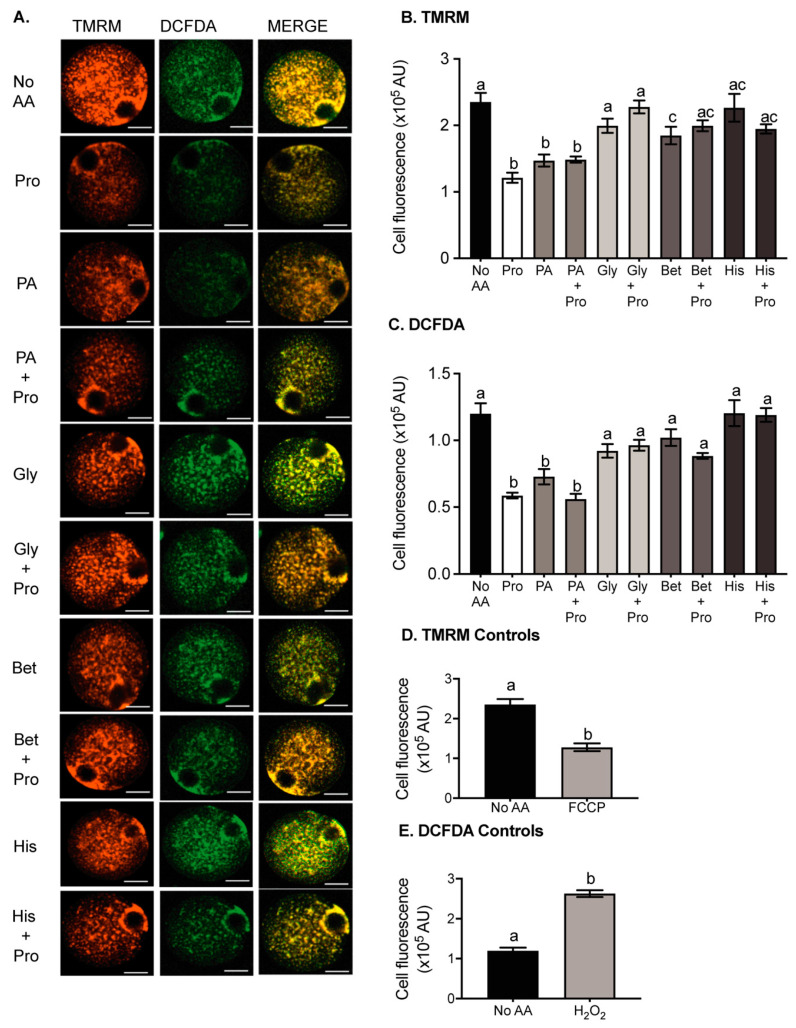
The effect of amino acids on ROS production and mitochondrial activity in oocytes. Oocytes were incubated for 3 h in medium containing either no amino acids (no AA) or 0.4 mM Pro or 0.4 mM PA, Bet, His, or Gly ± 0.4 mM Pro. Oocytes were then loaded with DCFDA for ROS, and TMRM for mitochondrial activity, and imaged by confocal microscopy. (**A**) Representative fluorescent images of oocytes in each condition. Quantified cell fluorescence for (**B**) TMRM and (**C**) DCFDA. Bars represent mean ± SEM for 28–71 oocytes, obtained from at least 3 independent experiments. Control oocytes stained with (**D**) TMRM and (**E**) DCFDA were transferred to a 10 µL medium containing 60 µM H_2_O_2_ or 5 µM carbonyl cyanide-p-trifluoromethoxyphenylhydrazone (FCCP) before imaging. Scale bar = 20 µm for all images. Data were compared by 1-way ANOVA with Tukey’s multiple comparisons test. Bars not sharing the same letter are significantly different (*p* < 0.05).

## Data Availability

All data is included in the published article. Data and materials will be made available upon request.

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
