# Peer review of "In Vitro Fertilisation of Mouse Oocytes in L-Proline and L-Pipecolic Acid Improves Subsequent Development"

_cells, 2021, doi:10.3390/cells10061352_

Round 1

Reviewer 1 Report

Manuscript „IVF of mouse oocytes in L-Proline and L-Pipecolic acid improves subsequent development” by Treleaven et al., is focused on the effect of presence of proline and pipecolic acid in media during IVF. Authors discovered that the presence of both improved subsequent embryonic development, which was demonstrated by increased proportion of blastocyst and also by increased number of cells in ICM. They showed that this effect was inhibited by presence of other amino acids in the media and was only partially dependent on sodium. Authors showed that the proline and pipecolic acid decreased the mitochondrial activity and reduced the amount of reactive oxygen species in oocytes.

This study is interesting and demonstrates that there are possibilities to improve embryonic development by relatively simple changes to IVF media. Given the relatively low development of embryos after IVF, especially in some species, this study is important for the field of reproductive biology.

I have only two questions:

  1. Would addition of proline and pipecolic acid during IVF had any effect also on the development of embryos cultured in AA media, for example KSOM +AA?
  2. Would it be possible to recapitulate the effect of proline and pipecolic acid on embryonic development (blastocyst rate + number of ICM) during ICSI as well?

Author Response

  1. Would addition of proline and pipecolic acid during IVF had any effect also on the development of embryos cultured in AA media, for example KSOM +AA?

Since the beneficial effect occurs when Pro or PA is present during IVF we predict that the positive effects on later embryo development would occur in any culture media, including those containing all AAs.

  1. Would it be possible to recapitulate the effect of proline and pipecolic acid on embryonic development (blastocyst rate + number of ICM) during ICSI as well?

ICSI has been shown to increase ROS in bovine oocytes (Ashibe et al. 2019, Theriogenology, 133: 71-78). Therefore, we postulate that it would be possible to recapitulate the effect of Pro and PA on embryonic development, if they were present during ICSI, due to their role in reducing ROS in oocytes.

Reviewer 2 Report

The presented manuscript entitled “IVF of mouse oocytes in L-Proline and L-Pipecolic acid improves subsequent development” by Treleaven et al. described the effect of selected amino acids present in the in vitro fertilization medium on subsequent pre-implantation embryo development in mouse.

This research field is quite interesting and worth to be explored, although what it is true for the mouse could not be true for human or other animals, but it is important as proof of concept that multiple amino acids could prevent the beneficial action of specific amino acids present in the culture medium and influencing subsequent embryo development. In addition, the effect is related to a diminished mitochondrial activity and ROS level, as already described in neuronal cells. The experiments are well organized and presented but the reviewer ask for some minor revisions, as follow:

-          In the Title, it is convenient to use the name in full, instead of the acronym “IVF”

-          The first sentence in the Introduction need to be accompanied by some bibliographic references

-          In Section 2.3 the procedure of sperm collection from the epididymis should be described

-          In section 3.1 the Authors should indicate at least the percentage of in vitro fertilization obtained, at the end of the sentence “The proportion of oocytes fertilised was not affected by any amino acid treatment (data not shown)”

-          The data presented in Figure 1 A-D show the percentages on “NoAA” and “Pro” repeated in the four panels. The reviewer suggests to pool the data differently, presenting a panel for each embryo stage (2C, 4C, Comp and BL), as in panel 1E and 1F for the embryo’s cell number

-          The Authors can discuss more on the trophectoderm cell number that is not the same as in vivo in none of the in vitro conditions tested. This could have effect later on, since these cells would contribute to the placental structure fundamental for the embryo growth after implantation.

-          check for some typo error, as in reference # 62

Author Response

  1. In the Title, it is convenient to use the name in full, instead of the acronym “IVF”

Thank you, we have updated the title to “In vitro fertilisation of mouse oocytes in L-Proline and L-Pipecolic acid improves subsequent development”.

  1. The first sentence in the Introduction need to be accompanied by some bibliographic references

We have added references to the introduction sentence (Line 33).

  1. In Section 2.3 the procedure of sperm collection from the epididymis should be described

The sperm collection procedure is now described (Line 127-130).

  1. In section 3.1 the Authors should indicate at least the percentage of in vitro fertilization obtained, at the end of the sentence “The proportion of oocytes fertilised was not affected by any amino acid treatment (data not shown)”

We have added the % fertilisation (section 3.1). “The proportion of oocytes fertilised was not significantly affected by any amino acid treatment (No AA 45%, Pro 47%, PA 50%, Gly 45%, His 39%, Pro + PA 45%, Pro + Gly 45%,  His + Pro 39%).” (Lines 208-210).

  1. The data presented in Figure 1 A-D show the percentages on “No AA” and “Pro” repeated in the four panels. The reviewer suggests to pool the data differently, presenting a panel for each embryo stage (2C, 4C, Comp and BL), as in panel 1E and 1F for the embryo’s cell number

Thank you for your suggestion. We have presented the data in this way because the experiments testing each amino acid were performed with the paired no AA and Pro as controls. In this way we were able to control for any variability between experiments and embryo isolations. We looked at pooling the data in stages, as you suggested, however this results in uneven replicate numbers and statistical analysis that is no longer based on the experimental design. The data in panel E&F were pooled because it is based on values from individual embryos.

  1. The Authors can discuss more on the trophectoderm cell number that is not the same as in vivo in none of the in vitro conditions tested. This could have effect later on, since these cells would contribute to the placental structure fundamental for the embryo growth after implantation.

Yes thank you for the suggestion, we have added more about the effect on trophectoderm cell number to discussion. “This is consistent with mice studies that have shown IVF reduces trophoblast cell numbers and affects subsequent placental development due to alterations in gene expression including the downregulation of SLC genes [81, 82]” (Lines 415-416).

  1. check for some typo error as in reference # 62

This has been corrected.

Reviewer 3 Report

The research article n°cells-1232308 titled “IVF od mouse oocytes in L-Proline and L-Pipecolic acid improves subsequent development” is very well written and the figures are clear and relevant. The aim of this publication is to analyze the effects of exposure of oocytes to L-Proline or its homologue L-Piperolic acid during in vitro fertilization on subsequent embryo development until the hatched blastocyst stage. The work described has been carried out on mouse oocytes following previous results of the same research group, showing beneficial effects of L-Proline or L-Glutamine addition on mouse embryo development. The results demonstrated that embryos fertilized in medium with proline and/or its homologue increased blastocyst formation and inner cell mass cell numbers compared to embryos fertilized in medium containing no amino acids, betaine, glycine or histidine. The beneficial effect of proline was prevented by addition of excess betaine, glycine and histidine, indicating competitive inhibition of transport-mediated uptake. This study has been completed by the analysis of proline transporter expression and uptake, as well as the evaluation of these amino acid effects on mitochondrial activity and reactive oxygen species production.

In my opinion, this article must be accepted after minor revisions as follows:

Figure 3: the significance of the puncta presence of the PROT transporter across the cytoplasm should be explained in the results. The immunolabelling of the Na+-dependent Pro transporter GLYT1 should be also shown, since this transporter seams to be more relevant for the Proline uptake, according to the discussion part (beginning of page 13).

Question 1 to be discussed: L-Pro is shown to have a beneficial effect during IVF (this article) and embryo culture (your previous article). Are both effects additive or not? Are the mechanisms under both effects the same?

Question 2 to be discussed: the redundancy of amino acid transporters expressed in oocyte seams to allow an adjustment of metabolism to the presence of substrate specificities, according to the discussion part (beginning of page 13). However, are the Na+-dependent and Na+-independent Pro transporters inducing the same effects on mitochondrial activity and ROS production, or these effects are transporter-dependent?

Question 3 to be discussed: some references described studies carried out on other mammalian species. Could the proline effects described in this study be applicable to other mammalian species? Do you know if the different type of Pro transporter are differentially expressed according to the stage of embryo development and/or the embryo species?

References to be completed: # 3 (article 140); #5 (article 705); # 6 (no reference, is-it personnel information?); # 26 (90(4):81 p.1-9); # 36 (article 340); # 51 (article 9263) and # 61 (article 279).

Author Response

  1. Figure 3: the significance of the puncta presence of the PROT transporter across the cytoplasm should be explained in the results. The immunolabelling of the Na+-dependent Pro transporter GLYT1 should be also shown, since this transporter seems to be more relevant for the Proline uptake, according to the discussion part (beginning of page 13)

Thank you for your suggestion. We have included a comment about the significance of the staining (line 265-267). “This indicates that PROT is involved in both membrane and vesicular transport of Pro. It is likely that expression of PROT in vesicles contributes to Pro accumulation and/or signalling processes important for oocyte competency.”

We have also included the following in the discussion to clarify why we didn’t further explore GLYT1 expression in the oocyte: “GLYT1 is located in the oocyte plasma membrane and actively transports Gly at fertilisation and throughout preimplantation development [1, 62, 67]. As GLYT1 substrates Gly and Sar both reduced Pro uptake we can’t rule out the possibility that GLYT1 takes up Pro and PA in oocytes. However its weak affinity for Pro makes it unlikely to be the main contributor [63]” Line 340-343.

  1. Question 1 to be discussed: L-Pro is shown to have a beneficial effect during IVF (this article) and embryo culture (your previous article). Are both effects additive or not? Are the mechanisms under both effects the same?

At this stage we do not know if the effects are additive or not. Our preliminary data suggests that yes the mechanisms underlying both effects are the same (unpublished data).

  1. Question 2 to be discussed: the redundancy of amino acid transporters expressed in oocyte seems to allow an adjustment of metabolism to the presence of substrate specificities, according to the discussion part (beginning of page 13). However, are the Na+-dependent and Na+- independent Pro transporters inducing the same effects on mitochondrial activity and ROS production, or these effects are transporter-dependent?

We believe that the effects on mitochondrial activity and ROS production are Pro-specific rather than transporter-dependant. The presence (and redundancy) of Pro transporters in the oocyte further highlights the physiological importance of Pro.

  1. Question 3 to be discussed: some references described studies carried out on other mammalian species. Could the proline effects described in this study be applicable to other mammalian species? Do you know if the different type of Pro transporter are differentially expressed according to the stage of embryo development and/or the embryo species?

Yes other Pro transporters have been identified at different stages of development. We have mentioned these in lines 328-330, 335-336 and 340-342 and added a sentence at line 332-333.  “Of these, SIT1 [34, 35], GLYT1 [1, 62, 63], B0AT2 [64], PROT [35] and SNAT2 [65] have been described in mouse embryo developmental stages after fertilisation.”

Pro transporters in other species haven’t been investigated. 

  1. References to be completed: # 3 (article 140); #5 (article 705); # 6 (no reference, is-it personnel information?); # 26 (90(4):81 p.1-9); # 36 (article 340); # 51 (article 9263) and # 61 (article 279).

All references have been corrected accordingly